# Potential Use of Antioxidant Compounds for the Treatment of Inflammatory Bowel Disease

**DOI:** 10.3390/ph16081150

**Published:** 2023-08-14

**Authors:** Alexander V. Blagov, Varvara A. Orekhova, Vasily N. Sukhorukov, Alexandra A. Melnichenko, Alexander N. Orekhov

**Affiliations:** 1Laboratory of Angiopathology, Institute of General Pathology and Pathophysiology, 8 Baltiiskaya Street, Moscow 125315, Russia; v.a.orekhova@yandex.ru (V.A.O.); vnsukhorukov@gmail.com (V.N.S.); sasha.melnichenko@gmail.com (A.A.M.); 2Institute for Atherosclerosis Research, Osennyaya Street 4-1-207, Moscow 121609, Russia

**Keywords:** IBD, ulcerative colitis, Crohn’s disease, oxidative stress, ROS

## Abstract

Since inflammatory bowel diseases (IBDs) are chronic, the development of new effective therapeutics to combat them does not lose relevance. Oxidative stress is one of the main pathological processes that determines the progression of IBD. In this regard, antioxidant therapy seems to be a promising approach. The role of oxidative stress in the development and progression of IBD is considered in detail in this review. The main cause of oxidative stress in IBD is an inadequate response of leukocytes to dysbiosis and food components in the intestine. Passage of immune cells through the intestinal barrier leads to increased ROS concentration and the pathological consequences of exposure to oxidative stress based on the development of inflammation and impaired intestinal permeability. To combat oxidative stress in IBD, several promising natural (curcumin, resveratrol, quercetin, and melatonin) and artificial antioxidants (N-acetylcysteine (NAC) and artificial superoxide dismutase (aSOD)) that had been shown to be effective in a number of clinical trials have been proposed. Their mechanisms of action on pathological events in IBD and clinical manifestations from their impact have been determined. The prospects for the use of other antioxidants that have not yet been tested in the treatment of IBD, but have the properties of potential therapeutic candidates, have been also considered.

## 1. Introduction

Inflammatory bowel disease (IBD) is characterized by recurrent episodes of inflammation of the gastrointestinal tract caused by an abnormal immune response to the intestinal microflora [1]. Inflammatory bowel disease includes two types of idiopathic bowel disease, which differ in the location and depth of damage to the intestinal wall. Ulcerative colitis (UC) involves diffuse inflammation of the lining of the colon. Ulcerative colitis most commonly affects the rectum (proctitis) but can spread to the sigmoid colon (proctosigmoiditis), and it can also spread out of the sigmoid colon (distal ulcerative colitis) or may cover the entire colon (pancolitis) [2]. Crohn’s disease (CD) is characterized by transmural lesions of any part of the gastrointestinal tract (GIT), with the most common pathologies occurring in the ileum and colon [2].

High mortality is typical for patients with Crohn’s disease, while the main causes of death are the direct progression of the disease itself, concomitant infections, complications from surgery, and multiple organ failure. IBD is one of the main risk factors for colorectal cancer [3]. At present, the etiology of IBD is not fully understood. Various causes of IBD have been considered, including smoking and diet, but none of them is dominant [4]. It is reliably known that the highest risk of developing IBD is in people who have mutations that determine their predisposition to IBD, the state of dysbacteriosis, and the disruption of the immune tolerance caused by it. Thus, a number of both hereditary and acquired factors are likely to be involved in the occurrence of IBD.

The average incidence of IBD is about 10 cases per 100,000 people in UC and about 7 cases per 100,000 people in CD [5]. The highest incidence of IBD occurs in the United States, Canada, and European countries, while developing countries in Asia and Africa show a lower incidence [5]. On an age scale, young people between the ages of 15 and 30 have the greatest risk of developing IBD. Crohn’s disease is somewhat more common to women than in men, but ulcerative colitis is equally present in both sexes [6].

The gut immune system plays a key role in the pathogenesis of IBD. The intestinal epithelium prevents the penetration of bacteria or antigens into the bloodstream due to hermetic intercellular connections. In IBD, these junctions are damaged either due to the disruption of the primary barrier functioning, or as a result of the development of inflammation. Additional defense mechanisms are based on the production of mucus by goblet cells and the production of antimicrobial proteins alpha-defensins by Paneth cells. Increasing inflammation causes even more damage to the structure of the epithelium, which leads to the spread of microbial populations in the intestinal wall, which in turn is a signal for a further increase in inflammation [1].

The basis of IBD treatment is the use of oral drugs based on 5-aminosalicylic acid (5-ASA) with the addition of steroids during an exacerbation of the disease and the transition to tumor necrosis factor inhibitors and immunomodulatory drugs in the absence of remission. Despite tremendous advances in IBD therapy in recent years, approximately 30% of patients do not respond to anti-TNFα therapy in the first place, and even among those who respond, up to 10% lose their response to the drug each year [7]. Additionally, the disadvantage of using current drugs for the treatment of IBD is the risk of developing severe side effects associated with the occurrence of infectious, as well as neoplastic processes [7]. Based on this, it becomes clear that the creation of new effective and safe drugs for the treatment of IBD is an important task for the pharmaceutical industry.

It is known that oxidative stress is a concomitant factor in the pathogenesis of various inflammatory diseases, since reactive oxygen species are direct modulators and initiators of the inflammatory response. Understanding the role of oxidative stress in the pathophysiology of IBD is important for evaluating the relevance of using antioxidants in the treatment of IBD. In most cases, antioxidants are natural compounds and do not cause serious side effects. In addition, the popularity of many of these compounds and methods due to their production from plant materials make the development of therapeutic agents based on them faster and the production less resource-intensive.

Increased oxidative stress with a pronounced weakening of antioxidant protection in IBD was identified in the 1990s [8,9]. This prompted the consideration of antioxidants as potential therapeutic agents in the treatment of CD and UC. Initially, such compounds were investigated in combination with already-registered drugs for the treatment of IBD. Thus, in one study [10], it was demonstrated that combination therapy with sulfasalazine or prednisolone with the addition of one of the antioxidants (allopurinol or dimethyl sulfoxide) led to a greater reduction in relapses among patients with UC than therapy with sulfasalazine or prednisolone alone. Better efficacy of allopurinol in combination with 5-ASA than 5-ASA alone during a 6-month but not a 12-month UC treatment period was shown in a study [11]. Thereby, the primary issues remain the long-term effectiveness of antioxidant therapy against IBD and the possibility of using antioxidants as monotherapy agents in the treatment of CD or UC.

## 2. The Role of Oxidative Stress in IBD

### 2.1. Mechanisms of the Formation of Reactive Oxygen Species in the Cell

ROS (reactive oxygen species) are chemical molecules containing one oxygen atom, which, as a result of cellular and extracellular reactions, becomes more reactive than oxygen itself. ROS include both free radical and non-free radical oxygen intermediates (peroxides), such as superoxide radicals (O_2_^−^), hydrogen peroxide (H_2_O_2_), hydroxyl radicals (OH), and singlet oxygen (1 O_2_) [12].

Oxidative stress is defined as an imbalance between the production of oxidants or ROS and their removal by defense mechanisms or antioxidants. Disruption of this redox balance can result in damage to important cellular components, including proteins, lipids, and DNA, with potential whole-body effects and an increased risk of mutagenesis. ROS are also involved in the initiation and development of pathological processes, including aging, cancer, insulin resistance, diabetes mellitus, cardiovascular disease, Alzheimer’s disease, etc. [13].

ROS are formed in two ways. The first is based on the redox reactions occurring in the ETC (electron transport chain) on the mitochondrial membrane during oxidative phosphorylation. ROS generated by this method are the main ROS concentrated in cells [14]. The second is associated with the protective response of immune cells to a bacterial infection or other pathological processes leading to inflammation [15]. These ROS play the role of signaling molecules in the body’s immune response.

During respiration, electrons donated from NADH in complex I and FADH2 in complex II pass to complex III through ubiquinone, then through cytochrome C to complex IV, and then the electrons are transferred to molecular oxygen with the formation of water. During oxidative phosphorylation, ETC protein complexes are involved in the process of creating a proton gradient, which is based on an increase in the concentration of protons in the mitochondrial intermembrane space and a decrease in their concentration in the matrix. As a result of the reverse flow of protons into the matrix, ATP is generated [16]. ROS, namely, superoxide, is a by-product of the reaction during electron transfer in the ETC to molecular oxygen [17]. Superoxide is produced by complex I and complex II into the mitochondrial matrix, while complex III additionally directs superoxide into the intermembrane space [18].

As noted, the second way to generate ROS is associated with a protective reaction of immune cells. In this case, ROS are produced by the enzyme nicotinamide adenine dinucleotide phosphate (NADPH) oxidase [15]. NADPH oxidase is associated with the cytoplasmic membrane and is a complex of proteins. The main stimuli that activate the work of NADPH oxidase are foreign microorganisms, increased inflammation, induction of calcium signaling, and an increase in the concentration of growth factors [14]. In addition to the superoxide radical, NADPH oxidase also produces hydrogen peroxide [19].

The tubular network of the endoplasmic reticulum (ER) has a unique oxidative environment, and during stressful conditions, redox signaling mediators are the main initiators of ROS production and directly affect protein folding and secretion [20]. The microsomal cytochrome P450-dependent monooxygenase system, whose role is associated with the metabolism of xenobiotics, is the dominant source of ROS production in hepatocytes [21].

Peroxisomes also contain ROS-producing enzymes, such as xanthine oxidase, which provides the formation of hydrogen peroxide [22]. The catalase enzyme is responsible for the neutralization of peroxide, catalyzing the decomposition of hydrogen peroxide into water and oxygen. The lysosomal electron transport chain, which facilitates the movement of protons to maintain an optimal pH for acid hydrolases, generates OH radicals [16].

### 2.2. Development of Oxidative Stress in IBD

In the human body, the gastrointestinal tract (GIT) contains a significant number of both stimuli and sources leading to the production of ROS. The epithelial layer, which performs a protective function, cannot provide complete protection against irritants, which include absorbed food components and those coming from outside, as well as microorganisms living in the intestine itself. It can cause an inflammatory reaction leading to the activation of neutrophils and macrophages that produce inflammatory factors, including ROS, which further enhance inflammation [23].

Impaired intestinal barrier function, leading to increased intestinal permeability, is a hallmark of the pathogenesis of IBD. Increased intestinal permeability is determined by the disruption of tight junctions, which leads to the release of pro-inflammatory agents, including ROS, which contribute to the progression of the pathological cascade [24]. As described above, the main ROS producers in IBD are macrophages and neutrophils. With increased inflammation, increased infiltration of these types of immune cells through the intestinal mucosa was noted, accompanied by the release of a large amount of ROS, which in turn play the role of attractants for immune cells [25]. Increased production of ROS in the inflammatory microenvironment of the intestine contributes to enterocyte damage caused by the ability of ROS to cause the destruction of biological macromolecules: proteins, lipids, and DNA. Cell damage leads to the release of cellular components into the extracellular space, where they can act as DAMPs, causing an additional increase in inflammation, including increased production of tumor necrosis factor (TNF)-α and interleukins IL-6 and IL-1β [26]. Dendritic cells play an important role, recognizing the formed DAMPs and then activating naive T-cells, which stimulate their proliferation and maturation and lead to the development of an adaptive immune response [27]. Both main groups of T-lymphocytes take part in IBD, namely, CD8+ T-lymphocytes and CD4+ T-lymphocytes, and the latter can be conditionally divided into two subgroups: effector and regulatory. If effector T-lymphocytes contribute to the further development of inflammation, then regulatory T-cells weaken it. The degree of development of IBD depends, in particular, on the ratio of these cell types [27].

Along with a direct role in the pathogenesis of IBD, oxidative damage to macromolecules, such as DNA damage and actin carbonylation with tubulin nitration, can be initiating factors in the development of colorectal cancer [28]. The relationship of IBD with cancer can also manifest itself when common signaling pathways are activated, including those associated with increased production of the p53 oncogene. Activation of p53 is closely associated with the development of oxidative stress through the expression of the p85 protein, which is a signal molecule in ROS-related p53-dependent apoptosis [29].

A noteworthy fact is the change in the concentration of cellular antioxidant enzymes in the gut mucosa in IBD. The expression of most of the genes of these enzymes, including SOD and CAT, decreases with an increase in inflammation associated with the pathogenesis of IBD, while the expression of the GPx2 gene (a form of GPx specific to the gastrointestinal tract) increases [30]. In studies of tissues of UC patients, it was shown that GPx2 is located in the ER, where it is able to modulate COX-2 activity through the utilization of peroxides and, as a result, reduce the level of prostaglandin E2, which is one of the main mediators of inflammation [31]. Thus, GPx2 is one of the last lines of defense to curb inflammation and carcinogenesis in IBD. Among the pathological effects of ROS, specific for IBD, one can note the effect of nitric oxide (NO•) on chloride anions, which, as a result of such an effect, are released into the intercellular space. It leads to a decrease in the amount of water in the intestinal lumen, and results in the development of osmotic diarrhea [32]. A brief diagram showing the role of oxidative stress in IBD is shown in Figure 1.

## 3. Natural Antioxidant Cellular Protection

### 3.1. Superoxide Dismutase (SOD)

SOD is the most important enzyme protecting against O_2_^−^ radicals. The SOD enzyme family is named after the cofactors used to detoxify excess O_2_, such as Cu/Zn-SOD, Fe-SOD, Ni-SOD, and Mn-SOD. SOD is involved in the transformation of the superoxide radical into oxygen and hydrogen peroxide [33]. Thus, SOD can act as an initial defense against ROS as a result of rapid initiation under oxidative stress. Cu/Zn-SOD is presented in a dimeric form and is located in the cytosol, where Cu and Zn are associated with two subunits [34].

It is now well known that mitochondria are the main producers of ROS, as well as the main targets of ROS. Massive accumulation of ROS and free radicals in mitochondria is the reason for the induction of increased expression of Mn-SOD, which suppress oxidative damage in mitochondria. The presence of specific SOD isoforms in different subcellular compartments ensures control of the ROS level in the cell [35].

### 3.2. Catalase (CAT)

The molecular structure of CAT and its function are based on iron atoms in the composition of the enzyme molecule, which form four heme groups around themselves [33]. The main function of CAT is to clean cells from hydrogen peroxide, which has toxic properties. Peroxisomes are the predominant localization site of CAT in the cell. The rate at which CAT converts hydrogen peroxide into molecular oxygen and water is 6 million hydrogen peroxide molecules per minute. CAT is highly active in hepatocytes, kidney cells, and erythrocytes [36].

### 3.3. Glutathione Peroxidase (GSH-Px)

The main function of GSH-Px is associated with the protection of the cell under oxidative stress and under normal physiological conditions from hydrogen peroxide, reducing it to water, while simultaneously reducing lipid peroxide. Another extraordinary role of GSH-Px is associated with the inhibition of tumor growth as a result of modulation of the signaling pathways of lipoxygenase and cyclooxygenase [37]. In the human body, there are eight types of GSH-Px, which can be in the form of selenoproteins, which are associated with selenium. Each of the GSH-Px types has a predominant localization in a certain tissue, where it performs its antioxidant function by reducing hydrogen peroxide and other organic peroxides [36].

Thus, GSH-Px1 or cellular GSH-Px (cGSH-Px) is localized in all cell types. GSH-Px2 or gastrointestinal GSH-Px (GSH-Px-GI) is present in human organs such as the stomach, liver, and intestines, but it is not expressed in the heart and lungs [38]. GSH-Px3 or plasma GSH-Px (pGSH-Px) is expressed in the kidney, and more specifically in proximal tubular epithelial cells. GSH-Px4 is a phospholipid localized in various cell types. Its key feature is the ability to reduce phospholipid hydroperoxides to alcohols, thus protecting cell membranes from peroxidation, which is one of the main consequences of oxidative stress leading to cell death [39].

### 3.4. Glutathione Reductase (GR)

GR promotes an increase in the stores of reduced glutathione, which is the most abundant thiol compound in most cells. Reduced glutathione is directly involved in the control of oxidative stress, being a GSH-Px cofactor. The catalytic function of GR is based on the transformation of glutathione disulfide (GSSG) into reduced GSH via the energy of reduced NADP [40].

### 3.5. Thioredoxin (Trx)

The Trx antioxidant system, consisting of NADP, thioredoxin reductase (TrxR), and Trx, is very important in combating oxidative stress as an endogenous antioxidant system. Trx antioxidants are involved in DNA and protein repair by reducing ribonucleotide reductase and methionine sulfoxide reductase [41]. Trx systems in cells contribute to the reduction of oxidative stress in the cell due to the presence of thiol and selenol groups. Trx, together with TBP2 and ASK1 proteins, may be involved in the modulation of cell apoptosis, as well as regulate lipid and carbohydrate metabolism. Both the GSH system and the Trx system can protect against oxidative stress by efficiently removing various ROS [42]. Cytosolic Trx1 and mitochondrial Trx2 are major redox proteins that regulate cell proliferation and viability. The reduced/dithiol form of Trxs binds to ASK1 and inhibits its activity, causing blockage of apoptosis. When Trx is oxidized, it dissociates from ASK1, and apoptosis is launched [43].

### 3.6. Cellular Antioxidant Regulation

The transcription factor nuclear erythroid factor 2 (NRF2) is one of the main modulators of the redox state of the cell, regulating the expression of antioxidant enzymes. The resulting oxidative stress is the reason for the activation of NRF2, which leads to the activation of the expression of heme oxygenase-1, which is an important cytoprotective protein [44]. Normally, NRF2 is associated with Kelch-like ECH-associated protein 1 (KEAP1). ROS at elevated concentrations oxidize cysteine residues on KEAP1 in the NRF2–KEAP1 complex, which facilitates the detachment of KEAP1 from NRF2. Free NRF2 then moves to the nucleus, where it interacts with MAF, resulting in the formation of heterodimers required for binding to antioxidant-responsive elements (ARE) in the regulatory regions of several antioxidant genes that coordinate their expression [45]. The level of mitochondrial ROS is directly related to the likelihood of developing oxidative stress, and, therefore, to ensure adequate cellular homeostasis, a properly functioning system is required to maintain the concentration of ROS within acceptable limits. A sharp increase in the level of ROS as a result of oxidative phosphorylation is the cause of pathological processes in the cell, which lead to oxidative damage to cellular structures and the triggering of apoptosis. With a lack of antioxidant enzymes, namely, SOD2, Trx2, peroxiredoxin, and GPx, the clearance of ROS in the cell is reduced, and the redox balance is disrupted. In addition, the formation of superoxide near NO molecules is a high-risk factor for the appearance of the toxic radical peroxynitrite (ONOO−) [46].

## 4. Natural Antioxidants in the Treatment of IBD

### 4.1. Curcumin

Curcumin, isolated from the rhizomes of the *Curcuma longa* plant, has proven antioxidant and anti-inflammatory effects. Curcumin has been shown to be beneficial in animal models of IBD. Thus, it was found that curcumin caused a decrease in the DAI index, which determined the severity of the pathological process. Additionally, it reduces damage caused by inflammation and inhibits neutrophil infiltration. In addition, it causes a decrease in lipid peroxidation [15]. The mechanisms associated with its therapeutic activity are not yet fully understood; however, possible options can be considered, for example, inhibition of the activation of the transcription factors NF-κB and STAT3, and accordingly the subsequent suppression of the expression of proteins that play the role of pro-inflammatory factors, including ROS-producing enzymes such as COX-2 and iNOS [47]. Based on a number of studies, it was shown that the introduction of curcumin reduced the clinical symptoms of IBD and led to the increased proportion of patients who managed to achieve remission [48,49]. Its additional advantage is the absence of serious side effects [1]. A clinical study [50] examined the effect of combination therapy of mesalazine with modified curcumin, which had better hydrophilic properties in patients with mild to moderate UC. According to the results of the study, patients receiving combination therapy demonstrated endoscopic remission, as well as long-term clinical remission, which was assessed at 6 and 12 months, with remission rates of 95% and 84%, respectively.

### 4.2. Resveratrol

Resveratrol is a polyphenol found in grapes and other fruits. Studies in rat models of IBD have shown that resveratrol reduced the development of lipid peroxidation damage and caused an improvement in cellular antioxidant activity. It could be caused by an increase in glutathione peroxidase activity, as well as an anti-inflammatory effect by inhibiting the expression of pro-inflammatory factors, namely, IL-1β, IL-6, and TGF-β1, and it reduces fibrosis [51,52]. The antioxidant and anti-inflammatory effects of resveratrol have also been demonstrated in several clinical studies. Thus, it was found that resveratrol caused an increase in SOD activity and a decrease in the concentrations of C-reactive protein and TNF-α [52]. Resveratrol, like curcumin, lowers the DAI value [53]. A clinical study [54] demonstrated a significant decrease in the level of inflammatory markers TNF-α and hs-CRP, as well as a decrease in NF-κB activity in patients with mild to moderate UC after 6 weeks of resveratrol supplementation. The disadvantages of resveratrol include its low bioavailability due to rapid absorption in the gastrointestinal tract and liver, as well as poor solubility and low stability [32]. To overcome these shortcomings, a colonic delivery system was developed for resveratrol [55]. This system was made of composite nanoparticles consisting of zinc, pectin, and chitosan, inside which the active substance, resveratrol, was enclosed. Several variants of nanoparticles were obtained by changing a number of parameters, such as cross-linking pH, concentrations of components, cross-linking time, and drug concentration. The nanoparticles obtained at pH 1.5 with 1% chitosan, 2-hour cross-linking time, and a ratio of pectin to resveratrol of 3 to 1 had the best specificity for the colon. Another significant disadvantage of this compound is the possibility of toxicity to normal cells. In high concentrations, resveratrol cannot be used as a drug, as it is considered a toxic compound [56].

### 4.3. Quercetin

Quercetin has shown beneficial effects in animal models of IBD, including reduced weight loss, rectal bleeding, and bowel injury. In addition, quercetin has an antioxidant effect by initiating a decrease in the activity of the myeloperoxidase enzyme and an increase in the concentration of glutathione, as well as inhibition of lipid peroxidation. Its anti-inflammatory mechanism of action is the inhibition of TNF-α expression [57,58]. An important effect of quercetin regarding the course of IBD is to improve the composition of the intestinal microflora, in particular, by activating macrophage activity against a number of bacteria [57]. It was found using a mice model that 5 weeks of quercetin consumption led to the growth of Bacteroidetes and to a proportional decrease in Gram-negative proteobacteria and Gram-positive actinobacteria concentrations in the colon. Quercetin has also been shown to reduce intestinal permeability by activating tight junction repair [59]. Quercetin can be effectively used in combination with 5-aminosalacylic acid (5-ASA), which allows for a reduction in the dose of 5-ASA, thus minimizing the possibility of side effects [60].

### 4.4. Melatonin

Melatonin is an organic substance synthesized mainly in the pineal gland, as well as in organs such as the retina and ovaries, which have the ability to produce this natural hormone [61]. Compared to other antioxidants, melatonin has unique properties that prove its uniqueness compared to other antioxidants. First, it is soluble in both water and lipids. Due to this property, melatonin easily penetrates through cell membranes and reaches all cell compartments, but at the same time it is selective towards mitochondria. It is a relatively safe substance, the reported side effects of which are usually harmless [61]. Melatonin can bind heavy metals (for example, iron) and thus prevents the formation of ROS. In addition, melatonin enhances the mechanisms of antioxidant cell defense by activating antioxidant enzymes such as catalase, superoxide dismutase, and others. Another mechanism of melatonin action is based on the anti-inflammatory effect, as a result of which the balance of concentrations of pro-inflammatory and anti-inflammatory cytokines are shifted towards the second [62]. Melatonin has been shown to be effective in the treatment of UC as an adjunct to combination therapy with mesalazine [63]. In a clinical study [63], patients with UC who received adjuvant therapy with melatonin in addition to mesalazine maintained clinical remission for 12 months, and serum CRP levels remained stable. These results were very different from the group receiving mesalazine only: remission was not stable, and CRP levels increased. Another advantage of using melatonin in the treatment of IBD is its ability to modulate intestinal dysbiosis [64]. This study was conducted on two groups of mice with DSS-induced colitis: the first group did not receive any therapy, and the second received melatonin treatment. In the first group, in the plasma of mice, the antioxidant activity, measured by decolorization of ABTS radical cations, was lower than in the second. The proportions of intestinal bacterial communities also differed. In the first group, the percentages of the most common bacteria were as follows: Bacteroidetes (59%), Firmicutes (31%), and Proteobacteria (8%). In the second group, the most common bacterial group was Firmicutes (49%), followed by Bacteroidetes (41%); the amount of Proteobacteria did not change (8%).

## 5. Artificial Antioxidants in the Treatment of IBD

### 5.1. N-acetylcysteine

N-acetylcysteine (NAC) is a derivative of L-cysteine, one of the functions of which is associated with the inhibition of the formation of free radicals, the increase in the activity of antioxidant defense enzymes, and the suppression of the production of heat shock proteins, which are considered biomarkers of oxidative stress. NAC can be converted to cysteine, which is a substrate that is involved in the reduction of glutathione in the intestine [65]. NAC also has an anti-inflammatory effect by inhibiting the activation of the NF-kB signaling pathway, thus preventing increased expression of pro-inflammatory cytokines [66]. NAC also increases ATP levels, inhibits apoptosis by acting on caspases, and promotes proliferation, development, and regeneration of intestinal cells [67]. NAC may reduce intestinal barrier permeability by reacting with claudin and occludin proteins [67]. NAC also has a positive effect on the intestinal microflora [68]. In a model of chemically induced colitis in rats, the successful use of NAC in combination with mesalamine was shown, as a result of which the level of inflammatory factors such as iNOS, COX-2, and prostaglandin E2 decreased to a greater extent than when these compounds were administered alone [69]. In a clinical study [70], it was demonstrated that the use of NAC as an adjuvant therapy with simultaneous reduction in the dose of corticosteroids in UC led to an increase in the period of remission. Combination therapy with mesalazine and NAC, evaluated in a clinical trial [71], resulted in better clinical remission, accompanied by a decrease in pro-inflammatory chemokines in UC patients, in contrast to monotherapy with mesalazine.

### 5.2. Artificial Superoxide Dismutase

One of the enzymes with strong antioxidant properties is superoxide dismutase (SOD), which has been described in detail in the previous paragraphs. However, in diseases associated with high levels of oxidative stress, natural SOD is not effective, as it has a short half-life and is not stable in the gastrointestinal tract. Despite these obstacles, recombinant bacterial strains have been obtained that express human SOD, for example, a strain of *Lactobacillus fermentum*, which has shown an improvement in colitis symptoms and a decrease in mortality in mice [72]. Its antioxidant function is associated with blocking lipid peroxidation and reducing the concentration of MPO in the intestine. The inflammatory response in the gut is reduced as a result of decreased production of pro-inflammatory cytokines caused by activation of the transcription factor NF-ĸB. Clinical trials of the action of lecithinized superoxide dismutase (PC-SOD) were carried out in patients with ulcerative colitis [73]. This new SOD variant overcomes the shortcomings of the natural version of the enzyme, having a longer half-life, which contributes to its longer action in the intestine. PC-SOD administered intravenously at a dose of 40 or 80 mg per day for four weeks improved the clinical symptoms of IBD and decreased the DAI index. Additionally, according to the results of the study, minor side effects of PC-SOD were revealed, but their occurrence did not depend on the dose of the administered enzyme. Summarizing data on the use of the considered antioxidants for the treatment of IBD are presented in Table 1. Mechanisms of antioxidant action of described antioxidant compounds are shown in Figure 2.

## 6. Discussion

Based on the fact that traditional IBD treatment can cause a number of severe side effects, while the aforementioned natural and synthetic antioxidant compounds are well tolerated and have mild side effects in most cases, this therapeutic strategy is a promising avenue for IBD therapy. In addition, the effectiveness of antioxidants in the treatment of IBD also has been shown. The mechanism of antioxidant action is aimed at reducing overall inflammation through an effect on transcription factors, and in some cases, improving the composition of the intestinal microbiota and restoring tight junctions, which shows their high therapeutic specificity with respect to IBD pathogenesis. At the same time, additional larger-scale studies are required to determine the long-term effectiveness of antioxidant-based drugs, as well as to assess the possible risks of chronic toxicity. Determining the best dosage, optimal route of administration, and delivery form that overcome the clinical limitations of the antioxidant substances under consideration are additional tasks, which by solving, the question of the regular use of these compounds in clinical practice can receive a reliable answer. Animal studies are a valuable starting point for clinical trials in humans, but relatively few data have been obtained so far, including in the considered preclinical models of IBD. In view of the fact that IBD is a risk factor for the development of colon cancer, it is interesting to consider the possible antitumor effects of antioxidant compounds. Thus, in a number of studies [74,75], it was demonstrated that PAC ((3,5-bis(4-hydroxy-3-methoxybenzylidene)-N-methyl-4-piperidone)), which is a biological analog of curcumin, could activate apoptosis in cancer cells without being toxic to normal cells.

## 7. Conclusions

The use of antioxidants is a promising direction in the treatment of IBD, which, first of all, will reduce the frequency of severe side effects arising from the use of traditional therapies. Both natural and artificial antioxidants, in addition to directly affecting oxidative stress, are also able to reduce inflammation by regulating cytokine expression, affect the gut microbiota, and reduce tight junction damage. It has been observed that the most effective clinical use of antioxidants was in combination therapy with an already registered drug for the treatment of IBD, which gave a synergistic effect. However, more research is needed to gain a more complete understanding of the effects of antioxidants.

## 8. Future Perspectives

We propose that an important future step towards the use of antioxidants for the treatment of IBD is the identification of antioxidants with previously unseen efficacy in the treatment of IBD. Based on the shown therapeutic effects in relation to other inflammatory diseases, the use of such compounds in the therapy of CD and UC may have potential success. In one study [76] it was shown that rosemary extract significantly improved DAI in mice with DDS-induced colitis and helped restore the integrity of the intestinal barrier. The pharmacokinetics of the main components of rosemary extract was determined: carnosic acid (CA) and carnosol (CL). CA had an 11-fold higher serum concentration than carnosine but a shorter half-life: 3.5 h compared to 7.5 h. The revealed pharmacokinetics shows the potential regimens of the isolated compounds. CA should have a fast and powerful therapeutic effect; CL can be used as a drug with a slow but prolonged effect. It was also found that CL inhibits the expression of sestrin-2, the increased concentration of which was previously considered a favorable factor in certain diseases [77]. Thus, these studies also allow scientists to reveal new questions regarding the pathogenesis of IBD that have not been raised before. Another natural compound, anethole, which has shown early antioxidant and anti-inflammatory properties in several disease states, also has the potential to be used in the treatment of IBD. It reduced edema and penetration of immune cells into the focus of inflammation after it was administered to mice with acetic acid-induced colitis [78]. It also reduced malonaldehyde levels and gene expression of inflammatory mediators, including TNF-α, IL-1β, and TLR4.

First of all, researchers should pay attention to antioxidants that exhibit multiple effects and are considered the “strongest” representatives of antioxidants in their class, for example, ergothioneine, whose only antioxidant function has three options: preventing the formation of free radicals, binding ROS, and increasing the activity of natural antioxidants. Ergothioneine is able to inactivate singlet oxygen at a faster rate than other thiols. It has also been shown to be effective in a number of pathological conditions, including those associated with chronic inflammation [79]. Thus, compounds similar to ergothioneine, according to the described criteria, have great potential to become new drugs in the treatment of IBD.

## Figures and Tables

**Figure 1 pharmaceuticals-16-01150-f001:**
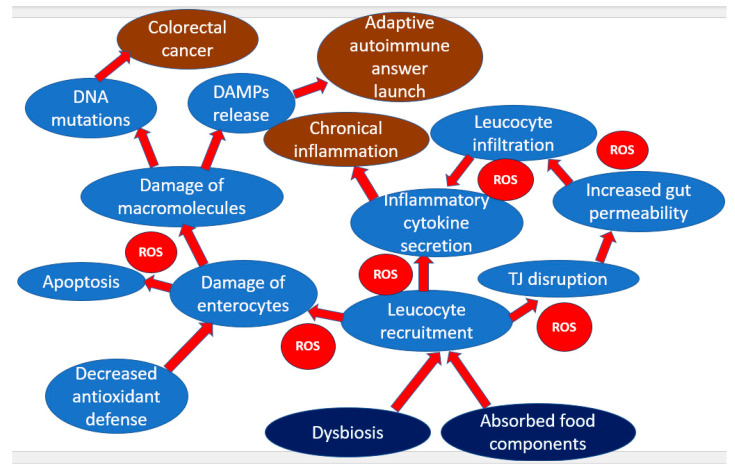
The role of oxidative stress in IBD.

**Figure 2 pharmaceuticals-16-01150-f002:**
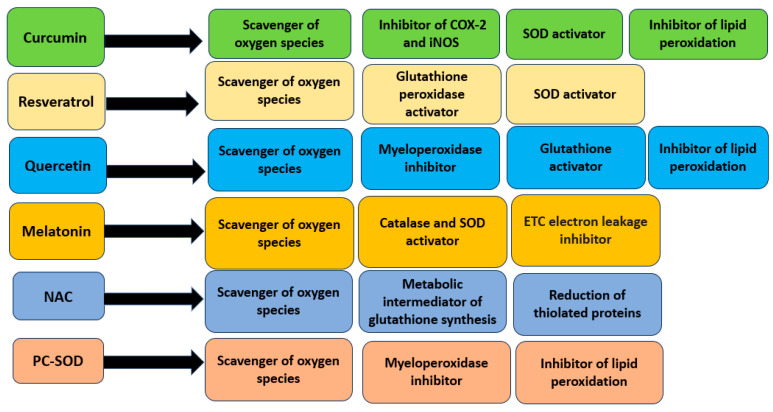
Mechanisms of antioxidant action of described antioxidant compounds.

**Table 1 pharmaceuticals-16-01150-t001:** Use of antioxidants for the treatment of IBD.

Compound	Chemical Structure	Mechanism of Action	Clinical Manifestation
Curcumin	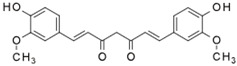	Inhibition of NF-κB and STAT3 pathways [47]. Inhibition of COX-2 and iNOS expression [47].	Reducing the clinical symptoms of IBD and increasing the achievement of remission [48,49].
Resveratrol	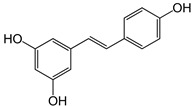	Increase in glutathione peroxidase and SOD activity [51,52]. Inhibition of NF-κB pathway [52].	Lowering the DAI value [53].
Quercetin	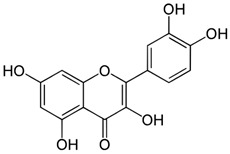	Decrease in the activity of myeloperoxidase [57,58]. Increase in the concentration of glutathione [58]. Inhibition of TNF-α expression [57]. Antibacterial activation of macrophages [57].	Reducing the clinical symptoms [60]. Improving the composition of intestinal microflora [57]. Reducing intestinal permeability [59]. Activating tight junction repair [59].
Melatonin	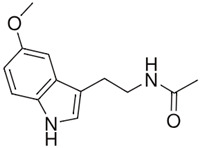	Increasing the activity of SOD and catalase [61]. Limiting the production of pro-inflammatory cytokines [62]. Enhancing the production of anti-inflammatory cytokines [62].	Reducing the clinical symptoms [63]. Modulation of intestinal dysbiosis [64].
N-acetylcysteine	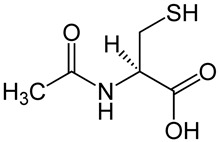	Inhibition of NF-κB pathway [66]. Inhibition of apoptosis by acting on caspases [67]. Promotion of intestinal cell proliferation [67]. Modulation claudin and occluding activity [67].	Reducing the clinical symptoms [69,70]. Improving the composition of the intestinal microflora [68]. Reducing intestinal permeability [68].
Artificial superoxide dismutase	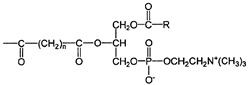	Direct inhibition of lipid peroxidation [72]. Decrease in the level of MPO [72]. Inhibition of NF-κB pathway [72].	Reducing the clinical symptoms of IBD [73]. Lowering the DAI value [73].

## Data Availability

Not applicable.

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
