# Peer review of "Potential Use of Antioxidant Compounds for the Treatment of Inflammatory Bowel Disease"

_pharmaceuticals, 2023, doi:10.3390/ph16081150_

Round 1
Reviewer 1 Report
As I found the main goal of the manuscript “Potential Use of Antioxidant Compounds for the Treatment of Inflammatory Bowel Disease” antioxidants in treatment of IBD. It is very interesting topic.
The topic is appropriate for this journal, however the paper contains fundamental flaws and should be rejected at this level or undergo extensive modification before publishing. My feedback on the manuscript is as follows:
Plagiarism exceeds 30%, indicating a serious problem.
The abstract is insufficient and too short.
The graphical abstract looks to be too simple; create visuals using AI technology.
The introduction is brief, but it might be expanded upon.
The second reference is absent from the text.
Figures should be used in the document to make it more useful and interesting.
Lines 186-190 are meaningless; please rewrite them to keep their meaning.
Add the chemical structures and MOAs of all antioxidants described in Section 4.
Figure one is unsuitable because it is again too simple
Table 1 does not include any references.
After the conclusion, include a future perspective.
The references do not as per the journal's author guidelines.
Lines 186-190 are meaningless; please rewrite them to keep their meaning.
Author Response
Response to Reviewer 1 Comments
Point 1: Plagiarism exceeds 30%, indicating a serious problem.
Response 1: The text was changed and added during editing, so this should improve the uniqueness. If the uniqueness is still low, then please send a plagiarism report where I can see the specific plagiarism in the text. Unfortunately, there is no time to rewrite the entire text.
Point 2: The abstract is insufficient and too short
Response 2: The abstract was expanded.
Point 3: The graphical abstract looks to be too simple; create visuals using AI technology.
Response 3: The graphical abstract was improved.
Point 4: The introduction is brief, but it might be expanded upon.
Response 4: It was expanded.
Point 5: The second reference is absent from the text.
Response 5: This reference was added.
Point 6: Lines 186-190 are meaningless; please rewrite them to keep their meaning.
Response 6: They were rewrited.
Point 7: Add the chemical structures and MOAs of all antioxidants described in Section 4.
Response 7: MOAs were shown in figure 2. The chemical structures were added in the second column of table 1.
Point 8: Figure one is unsuitable because it is again too simple
Response 8: Figure 1 was improved.
Point 9: Table 1 does not include any references.
Response 9: The references were added.
Point 10: After the conclusion, include a future perspective.
Response 10: It was done.
Point 11: The references do not as per the journal's author guidelines.
Response 11: The references were corrected.
Reviewer 2 Report
Brief summary:
This work reviews the current knowledge on therapeutic effects of certain antioxidant agents in Inflammatory Bowel Diseases (IBD). The work starts by presenting IBD for the general public, and continues by showing the role of oxidative stress in IBD. Last, the authors provide a comprehensive review of common antioxidant compounds and the observed or potential effects in IBD.
General concept comments:
In my opinion, the topic covered by this review is quite interesting, since the field of IBD have traditionally focus on blocking inflammatory pathways. Currently, there is the feeling in the field that a change of focus in needed to improve the treatment and care of patients with IBD. Oxidative stress plays an important role in the mechanisms of IBD, therefore antioxidants have potential to improve the treatment of IBD, either alone or in combination with current anti-inflammatory therapies. That is why in my opinion, the topic covered by this review is relevant.
The references added to this review are adequate, with one exception (commented in detail in “specific comments”). However, I think that authors could expand more the information from many of these references, detailing not only the main finding, but also the rationale and design of those studies. Also, in my opinion, authors could comment on many other antioxidant compounds, even when effects on IBD are still unknown, but there is a clear potential in IBD, given the effects on other diseases, for instance. This would improve the relevance of this review, since it would expand the gap of knowledge already identified by authors.
Specific comments:
Line 43: I miss references supporting the data on incidence. I know it is not the subject of the review, but a couple of references may improve the reliability of the data provided.
Line 44: there seems to be a grammatical mistake: the superlative form (“higher”) does not fit with the comparative form (“than”) used later in the sentence. The sentence can be like this: The highest incidence of IBD occurs in… while developing countries show a lower incidence. However, please, feel free to re-draft the sentence as your preference, as long as it is grammatically correct.
Line 62: in my opinion, when reporting such specific numbers, a reference is necessary, even though it is not the main topic of the review.
Line 79: abbreviations (ROS) must be explained (Reactive Oxygen Species) upon first citation.
Line 92: abbreviations (ETC) must be explained (Electron Transport Chain) upon first citation.
Line 102: please, specify it is the mitochondrial intermembrane space. Seems obvious for specialized authors, but readers out of the mitochondria or oxidative stress field might get lost.
Lines 128 to 133: this sentence is too long and readers might get lost. Please, consider re-structure this sentence by cutting it in two or three.
Line 163: at times, it gets confusing. I guess the authors are referring to expression of enzymes in the gut mucosa, by epithelial cells, but two paragraphs before the authors were talking about immune cells. Also, are authors referring to gene or protein expression? Please be more specific when mentioning expression.
Lines 170-173: please, revise this sentence. It is too long, full of “which”. Cutting long sentences in shorter ones makes the article more readable.
Lines 260-263: please, re-write this sentence.
Line 267: “such as COX-2 and iNOS” should go after “ROS-producing enzymes”. It would be still a very long sentence.
Line 268: here we found another grammatical mistake. Curcumin does not “participate” in studies. Curcumin (or the effects or curcumin) are studied. This sentence also needs to be re-written.
Line 270: Instead of “The clinical study” the sentence should start as “A clinical study”. This form is also found in other parts of the paper.
Line 274: The sentence leds to confusion. The clinical remission did not lasted 12 months. The referenced paper report that clinical remission was assessed at 6 and 12 months, with remission rates of 95% and 84%, respectively.
Line 276: Resveratrol is found in grapes, but not only in grapes, but also in other foods.
Lines 276-280: This sentence is too long and not well structured. Please, consider re-write it.
Line 290: I think it may be worth to expand the information regarding the findings in ref. 52.
Line 301: I think it is relevant to mention which particular bacteria are targeted by Quercetin. Please, expand information on ref. 28.
Line 350: “Despite with” is grammatically incorrect. Please, correct.
Line 306: I do not understand why melatonin has been classified as an artificial antioxidant. It is a natural compound, produced in the human body as authors detailed later.
Line 319: “are” shifted, instead of “is” shifted, because it is referring to the “concentrations” (plural). Please, correct.
Line 325: please, expand information regarding findings found in ref 62.
Lines 332-333: This statement needs a reference.
Lines 334-335: This statement also needs a reference.
Lines 335-336: This statement also needs a reference.
Line 337: Reference 63 is a review. I think authors should reference original sources, not other reviews.
Line 361: CDAI index was decreased, not increased.
Lines 361-364: As these lines are referencing Table 1 and act as a summary, I would move it to the start of the discussion.
Lines 371-372: There is a mistake in this sentence, please, re-write.
385-388: please, re-write this sentence.
Lines 389-392: I do not understand why the two referenced studies are not explained in more detail. Also, it may be worth reviewing antioxidants with potential therapeutic effect in IBD, even in the case of no existing data specifically on IBD. Authors could expand more on antioxidants compounds proving efficacy in other inflammatory diseases and focus on why efficacy in these other diseases may encourage other researchers to study these compounds in IBD.
Table 1: it would be useful for readers to add the references supporting the information detailed in this table.
The quality of english language is rather low. Authors should seek for professional assistance before re-submitting.
Author Response
Response to Reviewer 2 Comments
Point 1: Line 43: I miss references supporting the data on incidence. I know it is not the subject of the review, but a couple of references may improve the reliability of the data provided.
Response 1: The reference [5] was added.
Point 2: Line 44: there seems to be a grammatical mistake: the superlative form (“higher”) does not fit with the comparative form (“than”) used later in the sentence. The sentence can be like this: The highest incidence of IBD occurs in… while developing countries show a lower incidence. However, please, feel free to re-draft the sentence as your preference, as long as it is grammatically correct.
Response 2: It was corrected.
Point 3: Line 62: in my opinion, when reporting such specific numbers, a reference is necessary, even though it is not the main topic of the review.
Response 3: The reference [7] was added.
Point 4: Line 79: abbreviations (ROS) must be explained (Reactive Oxygen Species) upon first citation.
Response 4: It was done.
Point 5: Line 92: abbreviations (ETC) must be explained (Electron Transport Chain) upon first citation.
Response 5: It was done.
Point 6: Line 102: please, specify it is the mitochondrial intermembrane space. Seems obvious for specialized authors, but readers out of the mitochondria or oxidative stress field might get lost.
Response 6: It was done.
Point 7: Lines 128 to 133: this sentence is too long and readers might get lost. Please, consider re-structure this sentence by cutting it in two or three.
Response 7: It was done.
Point 8: Line 163: at times, it gets confusing. I guess the authors are referring to expression of enzymes in the gut mucosa, by epithelial cells, but two paragraphs before the authors were talking about immune cells. Also, are authors referring to gene or protein expression? Please be more specific when mentioning expression.
Response 8: It was clarified.
Point 9: Lines 170-173: please, revise this sentence. It is too long, full of “which”. Cutting long sentences in shorter ones makes the article more readable.
Response 9: It was done.
Point 10: Lines 260-263: please, re-write this sentence.
Response 10: It was done.
Point 11: Line 267: “such as COX-2 and iNOS” should go after “ROS-producing enzymes”. It would be still a very long sentence.
Response 11: It was corrected.
Point 12: Line 268: here we found another grammatical mistake. Curcumin does not “participate” in studies. Curcumin (or the effects or curcumin) are studied. This sentence also needs to be re-written.
Response 12: It was corrected.
Point 13: Line 270: Instead of “The clinical study” the sentence should start as “A clinical study”. This form is also found in other parts of the paper.
Response 13: It was changed.
Point 14: Line 274: The sentence leds to confusion. The clinical remission did not lasted 12 months. The referenced paper report that clinical remission was assessed at 6 and 12 months, with remission rates of 95% and 84%, respectively.
Response 14: It was corrected.
Point 15: Line 276: Resveratrol is found in grapes, but not only in grapes, but also in other foods.
Response 15: It was corrected.
Point 16: Lines 276-280: This sentence is too long and not well structured. Please, consider re-write it.
Response 16: It was corrected.
Point 17: Line 290: I think it may be worth to expand the information regarding the findings in ref. 52.
Response 17: It was described.
Point 18: Line 301: I think it is relevant to mention which particular bacteria are targeted by Quercetin. Please, expand information on ref. 28.
Response 18: It was described. The reference was specified.
Point 19: Line 350: “Despite with” is grammatically incorrect. Please, correct.
Response 19: It was corrected.
Point 20: Line 306: I do not understand why melatonin has been classified as an artificial antioxidant. It is a natural compound, produced in the human body as authors detailed later.
Response 20: The information about melatonin was transferred to the Section 4.
Point 21: Line 319: “are” shifted, instead of “is” shifted, because it is referring to the “concentrations” (plural). Please, correct.
Response 21: It was corrected.
Point 22: Line 325: please, expand information regarding findings found in ref 62.
Response 22: It was described.
Point 23: Lines 332-333: This statement needs a reference.
Response 23: The reference [69] was added.
Point 24: Lines 334-335: This statement also needs a reference.
Response 24: The reference [70] was added.
Point 25: Lines 335-336: This statement also needs a reference.
Response 25: The reference [70] was added.
Point 26: Line 337: Reference 63 is a review. I think authors should reference original sources, not other reviews.
Response 26: The reference [71] was added instead of [63].
Point 27: Line 361: CDAI index was decreased, not increased.
Response 27: It was corrected.
Point 28: Lines 361-364: As these lines are referencing Table 1 and act as a summary, I would move it to the start of the discussion.
Response 28: Do you mean this part of the text? “Also, according to the results of the study, minor side effects of PC-SOD were revealed, but their occurrence did not depend on the dose of the administered enzyme. Summarizing data on the use of the considered antioxidants for the treatment of IBD are presented in Table 1”. But it partially belongs to the previous study desription and separation of these text parts seems not good for the manuscript structure in my opinion.
Point 29: Lines 371-372: There is a mistake in this sentence, please, re-write.
Response 29: It was corrected.
Point 30: 385-388: please, re-write this sentence.
Response 30: It was corrected.
Point 31: Lines 389-392: I do not understand why the two referenced studies are not explained in more detail. Also, it may be worth reviewing antioxidants with potential therapeutic effect in IBD, even in the case of no existing data specifically on IBD. Authors could expand more on antioxidants compounds proving efficacy in other inflammatory diseases and focus on why efficacy in these other diseases may encourage other researchers to study these compounds in IBD.
Response 31: This issue was described in detail in new Section 8: Future perspectives.
Point 32: Table 1: it would be useful for readers to add the references supporting the information detailed in this table.
Response 32: It was done.
Point 33: The quality of english language is rather low. Authors should seek for professional assistance before re-submitting.
Response 33: The grammar mistakes were corrected.
Reviewer 3 Report
The authors have talked about IBD and focusing on ROS alone. As outcome antioxidant compounds are reported here in the manuscript and ROS mediates their mechanism. This manuscript seems to be a compilation of antioxidant compounds. IBD treatment includes the drugs mediated PGs and leukotrienes. I hope the exhaustive revision is required and possible correlations should be incorporated to benefit the readers.
Author Response
Response to Reviewer 3 Comments
Point 1: The authors have talked about IBD and focusing on ROS alone. As outcome antioxidant compounds are reported here in the manuscript and ROS mediates their mechanism. This manuscript seems to be a compilation of antioxidant compounds. IBD treatment includes the drugs mediated PGs and leukotrienes. I hope the exhaustive revision is required and possible correlations should be incorporated to benefit the readers.
Response 1: The manuscript has been significantly revised. Added a lot of information about conducted clinical studies.
Round 2
Reviewer 1 Report
Authors revised the manuscript substantially. May be accepted for publication in present form.
Reviewer 2 Report
The authors addressed all the issues previously detected.
Major grammar mistakes have been corrected. Also, very long sentences, which made the readability of the manuscript very difficult, have been cut in two. However, I think the way authors cut them in two, by replacing ", which..." by the form ". It..." is not the best solution. Although grammatically correct, better ways to re-write these sentences could have been found to improve the readability of the manuscript. Overall, although I could not find any major grammar mistake, I still think that the readability of the manuscript can be improved using a language editing service.
Reviewer 3 Report
The manuscript can be accepted.